# Osteoarthritis and Toll-Like Receptors: When Innate Immunity Meets Chondrocyte Apoptosis

**DOI:** 10.3390/biology9040065

**Published:** 2020-03-30

**Authors:** Goncalo Barreto, Mikko Manninen, Kari K. Eklund

**Affiliations:** 1Department of Rheumatology, Helsinki University and Helsinki University Hospital, 00014 Helsinki, Finland; kari.eklund@hus.fi; 2Translational Immunology Research Program, University of Helsinki, 00014 Helsinki, Finland; 3Orton Research Institute, 00280 Helsinki, Finland; mikko.manninen@orton.fi

**Keywords:** osteoarthritis, chondrocytes, toll-like receptors, apoptosis, innate immunity, cartilage

## Abstract

Osteoarthritis (OA) has long been viewed as a degenerative disease of cartilage, but accumulating evidence indicates that inflammation has a critical role in its pathogenesis. In particular, chondrocyte-mediated inflammatory responses triggered by the activation of innate immune receptors by alarmins (also known as danger signals) are thought to be involved. Thus, toll-like receptors (TLRs) and their signaling pathways are of particular interest. Recent reports suggest that among the TLR-induced innate immune responses, apoptosis is one of the critical events. Apoptosis is of particular importance, given that chondrocyte death is a dominant feature in OA. This review focuses on the role of TLR signaling in chondrocytes and the role of TLR activation in chondrocyte apoptosis. The functional relevance of TLR and TLR-triggered apoptosis in OA are discussed as well as their relevance as candidates for novel disease-modifying OA drugs (DMOADs).

## 1. Introduction: The Role of Immunity in OA

Clinical osteoarthritis (OA) is preceded by a preclinical stage, which, in conjunction with the presence of risk factors and/or other pathological factors, proceeds to the radiographic OA state. Numerous risk factors have been proposed as main pathogenic drivers associated with OA pathogenesis. Nevertheless, risk factors alone cannot explain OA pathogenesis. An emerging view proposes that risk factors, triggering mechanisms, and perhaps other known or unknown factors, act together, driving the disease into the radiographic stage.

A key role for inflammation has been established in OA, with the innate immune system being a major contributor to the inflammatory cycle of OA [1]. In OA, the immune system reacts to the mechanical, physiological, and biological changes in the joint over time. In contrast to the adaptive immune system, innate immunity plays an essential role not only in host defense against microbial agents but also in modulation of tissue homeostasis by recognizing distinct pathogen-associated molecular patterns (PAMPs) and damage-associated molecular patterns DAMPs, respectively, by pattern recognition receptors (PRR), such as toll-like receptors (TLR) and NOD-like receptors (NLR) [2]. Hence, traumatic injuries accumulated over time may lead to the release of cartilage DAMPs that may then activate a local innate immune system reaction [3]. Consequently, inflammatory pathways may be activated in resident cells and matrices leading to the upregulation of several cartilage matrices degrading proteases such as matrix metalloproteases (MMP)-1, MMP-3, MMP-13, and ADAMTS aggrecanases, while also downregulating aggrecan and collagen type II, a pattern generally seen in the OA chondrocytic phenotype. This gene’s expression patterns within articular chondrocytes are mediated via multiple intracellular pathways, such as the MAC, MAP kinases, and nuclear factor-κB pathways [4,5].

This interplay between mechanical traumas, environmental factors, potentiated by genetic predisposition and other risk factors, ensued by inflammation driven by an innate immune response and impaired cartilage repair, is one of the paradigm shifts of the OA pathogenesis theory [6].

## 2. Cartilage and Chondrocytes: The Hallmark of Osteoarthritis

The articular cartilage is a highly specialized tissue with unique biomechanical properties and solely populated by only one cell type, the chondrocyte, in an avascular, alymphatic, and aneural microenvironment [7]. Such uncommon characteristics are likely contributors for failed cartilage natural regeneration and the main challenges of cartilage reconstruction and engineering efforts [8].

The primary threat for cartilage matrix integrity is the disruption of the collagen network and proteoglycan by matrix-degrading proteases [9]. Cleavage and degradation of the matrix molecular components compromise the structure of the residing supramolecular proteins, which give cartilage tissue so unique properties. The enzymatic degradation of articular cartilage is the erosion of the pericellular matrix and, eventually, the interterritorial matrix will ultimately compromise and alter cartilage biomechanical properties leading to the destruction of articular cartilage [10].

The majority of proteases are localized in articular cartilage and are produced locally by chondrocytes, the only cell type present in articular cartilage. Moreover, chondrocytes act in concert to preserve the structural integrity of the extracellular matrix (ECM) in cartilage, making them central players of cartilage tissue homeostasis. Chondrocytes not only play a role in matrix catabolism, but they also actively regulate matrix anabolism. In healthy cartilage, chondrocytes synthesize new ECM molecules to replace damaged molecules; meanwhile, in OA, their anabolic activity is impaired and limited.

To date, extensive knowledge has been gathered about the degradation processes of the two primary components of articular cartilage: the collagen network and the enrooted proteoglycans. During early cartilage degeneration in OA, the most striking feature is the loss of aggrecan and bounded cationic proteoglycans [11]. Similarly noticeable, collagen content is slightly reduced; however, its network, in turn, is highly disrupted, making it also one of the key features of cartilage osteoarthritic changes. Importantly, the two have reciprocal effects, i.e., collagen network degradation leads to loss of bounded proteoglycans and proteoglycan loss alters the matrix biomechanical properties leading to cartilage overload, which will inflict further damage to the collagen network structure. 

Throughout the osteoarthritic cartilage, abnormal levels of many proteinases, including matrix metalloproteinases (MMPs), as well as members of the ADAM (a disintegrin and metalloproteinase) and ADAMTS (a disintegrin and metalloproteinase with thrombospondin type 1 motif) families, are associated with the increased matrix degradation in OA cartilage [12]. These enzymes play a significant role in cartilage degradation. However, continuous research efforts to identify the most crucial protease have been unsuccessful [12]. This may imply that many or all of them are vital, and hence, treatment strategies should target their upstream regulators.

Under pro-inflammatory conditions, the catabolic events are not sufficiently counterbalanced by the anabolic events given the impaired synthesis of cartilage matrix molecules, e.g., collagen type II, V, and aggrecan [13]. This continuing imbalance leads ultimately to the failure to compensate matrix cartilage damage induced in the local synovial joint (Figure 1).

## 3. TLRs in Osteoarthritis

TLRs are typical type I integral membrane receptors composed of a ligand recognition ectodomain, a single transmembrane helix, and a cytoplasmic signaling domain projecting from the inside part of the membrane [14]. Toll IL-1 receptor (TIR) domains are the signaling domain of TLR. The name originated since they are homologs to the IL-1R family members signaling domains [15]. 

As mentioned earlier TLRs have a critical role in the activation of innate host defense, particularly against infections by recognizing PAMPs, and tissue remodeling generated alarmins, also known as DAMPs. PAMP/DAMPS-TLR ligand recognition leads to the initiation of signaling cascades culminating in cellular activation. Currently, 11 TLR gene members have been discovered, numbered from 1 to 11, of which the first 10 are also functional in humans, albeit to date no natural ligand to TLR10 is known. TLR1, TRL2, TLR4, TLR5, TLR6, and TLR10 recognize microbial surface patterns and are therefore located on the cell membrane surface to enable an immediate response. Some other TLRs, such as TLR3, TLR7, TLR8, and TLR9, are expressed in the endosomes or phagosomes inner membranes to allow contact with internal microbial structures revealed upon microbial degradation/lyses, such as double- and single-stranded RNA and DNA (Figure 2).

TLR activation leads to host responses in the form of de novo expression of genes, such as inflammatory cytokines, IL-1, IL-6, IL-8, IL-12, tumor necrosis factor (TNF)-α, and cell-membrane-bound co-stimulatory molecules, such as intercellular adhesion molecule-1 (ICAM-1), and its counterpart, lymphocyte function-associated antigen-1 (LFA-1) [16]. 

In recent years, TLRs expression and signaling have been associated with OA pathogenic mechanisms. Moreover, it is important to study the TLRs local role in joint tissues, since OA is a disease of the entire joint.

Synovial inflammation (synovitis) is present in about half of the patients with OA and has been shown to correlate with cartilage damage severity [17]. TLR 1-7 and TLR9 expression is upregulated in the synovium of OA patients [18,19]. Moreover, functional studies have demonstrated that primary FLS from healthy and OA patients express TLRs, which actively respond to several DAMP ligands, such as tenascin-C or S100A8/A9 [20,21]. In the OA synovial fluid TLR4 is also present in the soluble form, and it is associated with the OA severity, making sTLR4 an intriguing biomarker [22,23]. TLRs activation and the ensuing NF-kB activation is followed by the production of chemokines (e.g., IL-8 and CCL5) and cytokines (e.g., IL-1, IL-6, and TNF) which participate in the recruitment of macrophages, granulocytes, and lymphocytes cell into the synovium of OA patients [24]. In the synovium, synovial fibroblasts and synovium lining FLS cells once activated lead to the secretion of cytokines and chemokines, which may cross-talk with cartilage and subchondral bone cells.

Mechanical loading and cross-talk between cartilage and the intimately connected subchondral bone make it possible that the chondrocytes secrete catabolic, inflammatory, and anabolic factors able to cross-talk with the subchondral bone microenvironment and vice versa (60,198). Interestingly, osteoblasts express TLR2 and TLR4 and can down-modulate cytokine secretion during chronic TLR4 challenge [25]. Besides, TLR4 activation can also modulate osteoclasts formation, survival, and activity [26]. This is of particular importance, given that reduced bone mineral density is a risk factor for OA development [27]. 

In joint trauma and OA, the synovial joint fluids and tissues contain increased concentrations of several DAMPs capable of activating TLRs. These include danger signals such as S100/calgranulin family (e.g., S100A4, the S100A8/A9 heterodimer, S100A11, S100A12), tenascin-C (TN-C), high- and low-molecular-weight hyaluronan, fibronectin isoforms, and SLRPs such as biglycan and decorin [20,21,28,29,30,31,32,33,34,35]. The majority of OA DAMPs are cartilage matrix derived, albeit some arise from cell apoptosis, e.g., high-mobility group box protein 1 (HMGB1) [36,37].

### TLRs and Chondrocytes

As cartilage biology is a central feature of OA, the local innate immunity regulation by TLRs in chondrocytes is therefore of particular interest. 

All TLRs are present in articular cartilage and are known to be upregulated in OA cartilage, particularly in the lower limbs such as knee and hip cartilage [38]. However, a recent study indicated that there are also joint-specific TLR expression patterns, particularly between hand and knee OA, which might also implicate differential immune mechanisms of large and small joints in OA [23].

Several DAMPs such as low-molecular-weight hyaluronan, S100A8/9, fibronectin fragments, and others, all with cartilage matrix origins are recognized by TLRs expressed in chondrocytes.

In line with cartilage aging contributing to OA pathogenesis, the advanced glycation end product (AGE), a known member of the DAMPS family, increasingly accumulates in articular cartilage and undermines collagen network mechanical properties [39]. Such an increase of accumulation, in turn, is recognized by PRRs such as RAGE and TLR4 expressed in human chondrocytes leading to the response of secretion of catabolic factors such as IL-6, COX-2, HMGB1, and MMP-13. Interestingly AGE stimulation leads to upregulated TLR4 expression levels [40]. Together this suggests PRRs, and, in particular, TLR4 to be involved in age-related cartilage degeneration in OA.

Fibronectin (FN) is a glycoprotein known to act as DAMP, which is upregulated in synovial fluid and cartilage of OA patients [41,42]. FN domains are able to induce catabolic responses on the cartilage [43]. Moreover, OA joint proteases participating in cartilage degradation were shown to cleave FN into different fragments lengths similar to those found in OA cartilage [44]. With the discovery of human TLRs, researchers were able to decipher how FN fragments are recognized. Some FN fragments have been shown to elicit catabolic responses through TLR2 recognition, while the domain 13-14 of 29 kDa FN fragments are recognized by TLR4 [45,46].

Serum amyloid A (SAA), a TLR4 ligand, is also found at increased concentrations as OA severity increases. SAA is upregulated in the serum and synovium of OA patients, and SAA levels correlate with the radiographic progression of OA. In vitro studies have demonstrated that SAA is recognized by TLR4 in human chondrocytes and synovial fibroblast and ensuing production of cytokines and collagenases [47].

Other known TLR DAMPs present at increased levels in the OA synovial joints are members of the calcium-binding proteins family, specifically, S100A8 and S100A9. S100A8 and S100A9, whose levels are also increased in OA cartilage lesions and correlate with the expression of MMPs and proteoglycan depletion [48,49]. Chondrocytes can synthesize S100A8 and S100A9, and their synthesis is upregulated by known OA inflammatory factors such as IL-1β, TNFα, IL-17, and IFN [48]. Furthermore, S100A8 and S100A9 are recognized by TLR4^+^ chondrocytes, leading to upregulated secretion of collagenases and aggrecanases and inhibition of collagen type II and aggrecan [49]. Moreover, a similar biological response is observed in S100A8/9-stimulated OA mouse models [50]. Interestingly, S100A8 and S100A9 levels in early symptomatic OA patients can predict osteophyte formation after two or five years [51].

Biglycan (BGN) and decorin (DCN) are two small structural proteoglycans with leucine-rich repeats (small leucine-rich proteoglycan (SLRP)) essential to the cartilage matrix structure [52]. However, several studies have demonstrated the BGN and DCN upregulation in OA cartilage and increased forms of BGN and the presence of DCN auto-antibodies in SF from OA [53,54]. Importantly, BGN is upregulated in OA SF, and BGN and DCN activate chondrocyte pro-inflammatory response, via TLR4, resulting in the upregulation of OA inflammatory factors such as IL-6, IL-8, MMPs, and nitric oxide levels [55]. Moreover, BGN was shown to induce cartilage degradation in cartilage explants models [55]. A list of the known receptors expressed in the synovial joints and their known DAMPS and modulators present in the OA joint is summarized in Table 1.

Finally, it is interesting to note that TLR4 expression seems to be regulated by the applied shear stress to chondrocytes, with high shear stress causing TLR4 upregulation. In contrast, prolonged shear stress leads to the downregulation of TLR4 and a TLR4 expression dependent inflammatory response [56].

## 4. TLRs and Chondrocyte Apoptosis

At the core of the failed regeneration and remodeling of degenerated cartilage might be the reduced number of chondrocytes in aged articular cartilage. As early OA progresses complex chondron start to form, also known as chondrocytes cluster or clones, as a result of increased metabolic activity and initial chondrocytic proliferation [11,63]. Eventually, the initial cell proliferation is counterbalanced by the failure to prevent the catabolic degeneration and reduced anabolic secretion leading to chondrocyte death, hypocellularity, and empty lacunas [11].

It is important to note that besides apoptosis other mechanisms of cell death, such as autophagy and necrosis, may contribute to the chondrocyte death but these mechanisms are outside the focus of this review.

### 4.1. The Interplay Between TLRs and Apoptosis of Chondrocytes in OA

#### 4.1.1. TLRs and Apoptosis

TLR signaling is dependent on five TIR domain-containing adaptors; however, only MyD88 and TRIF function as transducers [64]. TRAM and TIRAP are bridging adaptors, while SRAM regulates (negatively) TRIF. TLR activation via Myd88 (TLR2, TLR7, TLR8, and TLR9), TRIF (TLR3), or both (TLR4) can lead to apoptosis, demonstrating that Myd88 and TRIF can activate apoptosis independently and, perhaps, the cross-talk between the two signaling pathways (TLR4) can lead to apoptosis as well [65,66,67]. The Myd88-dependent pathway initiates the recruitment of MyD88 and TIRAP via the TIRTIR homophilic interaction as a result of TLR dimerization. Myd88 then forms a complex with IL-1R-associated kinases (IRAK4 and IRAK1) and TRAF6. TRAF6 activation is repercussioned in the TGF-β-activated kinase 1 (TAK1)–TAB1/2/3 complex leading to the activation of the canonical NF-kB pathway and the MAP kinases (Figure 2). Overexpression of Myd88 induces a low level of apoptosis [65]. However, TLR-MyD88 role in cell death is suggested to be mostly indirect, via the signaling induction of pro-apoptotic molecules such as TNF-α or NO [68,69].

Interestingly, TLR4 may move from the plasma membrane to the endosomes in order to switch signaling from MYD88 to TRIF. Both TLR3 and TLR4 recruit TRIF directly or through TRAM. The C-terminal region of TRIF, in turn, recruits the receptor-interacting protein 1 (RIP1), a serine/threonine kinase that can promote apoptosis via the recruitment of FADD and ensuing caspase-8 activation [70,71]. On the other hand, the N-terminal region of TRIF can recruit TRAF6 leading to the activation of NF-KB and the ensuing production of pro-apoptotic molecules. TRIF N-terminal can also bind to TRAF3, which, in turn, triggers the phosphorylation of IRF3 and the secretion of IFN-b. Interestingly, independently of TLR activation TRIF overexpression in macrophages and fibroblasts is sufficient to trigger apoptosis, while TRIF inhibition protects murine macrophages against poly(I:C)- and LPS-induced apoptosis [67,72].

#### 4.1.2. Chondrocyte Apoptosis Mediated by TLRs in Osteoarthritis

At this point, it is now evident that TLRs and apoptosis are two crucial players of the pathomolecular mechanisms of cartilage degeneration in OA. However, much remains to be understood regarding the evidence and role for TLR-induced chondrocyte apoptosis mechanisms in osteoarthritis.

From the biomechanical perspective, it is well known that altered joint biomechanics, such as high fluid shear, can induce the production of pro-inflammatory mediators like IL-1β, TNFα, and IL-6 by chondrocytes [73]. Importantly, a recent study demonstrated the TLR4 signaling pathway as the primary responsible mediator of pro-inflammatory response, where the duration of shear stress was able to regulate TLR4 activation and consequent cells senescence, death, and apoptosis, in a lipocalin (L)-type prostaglandin-dependent manner [56]. Importantly, an ex vivo (cartilage explant) model of post-traumatic OA has demonstrated that mechanical strain leads to the pro-inflammatory response, mediated by TLR 3, -7, and -9, and upregulated chondrocyte apoptosis in a dose-dependent mechanical strain effect [74]. Interestingly, TLRs expression is increased in OA cartilage, with notably higher expression at the cartilage surface zone where shear stress and strain forces take place [75]. Together, these results suggest that TLR4 may be an essential mediator between mechanical overload stress and the transduction to an inflammatory response.

A typical inducer of chondrocyte apoptosis is the pro-inflammatory environment of SF in OA. In OA, the SF has increasing levels of several DAMPS, such as tenascin, biglycan, decorin, fibronectin, and pro-apoptotic molecules, such as TNF-α and IFN [76]. As mentioned, TNF-α and IFN are known to cause apoptosis via direct TLR or indirect activation of TLR signaling cascades, hence, their upregulated production by OA joint cells may influence cell death levels observed in cartilage and other tissues. 

As mentioned earlier, TLR4 downstream activation leads to an increase in the production of nitric oxide (NO), several pro-inflammatory cytokines, and adipokines, which, together, mediate cartilage degradation. NO, is a highly reactive gas, known to participate in the molecular mechanisms of arthritic conditions. In chondrocytes from RA and OA joints, upregulated NO production levels are observed in contrast to healthy chondrocytes [77,78]. Strikingly, chondrocyte production of NO is higher than the one observed by macrophages, making chondrocyte one of the significant sources of upregulated NO amounts in the intra-articular space [79]. The TLR-NF-kB pathway induces downstream activation of nitric oxide synthase (NOS2) expression, which, in turn, leads to an increase in the production of NO. Moreover, the increased amount of NO is accompanied by increased chondrocytes apoptosis is probably due to the ability of NO to induce chondrocyte apoptosis but also catabolic and inflammatory responses [80,81]. Interestingly, not only TLR activation leads to NO production but also vice versa. Recent work demonstrated the ability of NO to upregulate TLR4-mediated neutrophil gelatinase-associated lipocalin (NGAL) production, a known catabolic adipokine implicated in cartilage degradation and associated with OA [82,83].

Another recent study demonstrated that amyloid TTR, present in synovial fluid and deposited in the cartilage surface of OA joints, may induce apoptotic cell death, and pro-inflammatory response in a partly TLR4-dependent manner [84].

Obesity, a known risk factor for OA, is associated with a state of low-grade inflammation and increased circulating levels of adipokines and free fatty acids (FFAs). Interestingly, adipokines/FFAs such as palmitate induces a TLR4-signaling dependent activation of caspases and cell death in IL-1β-stimulated normal chondrocytes. In an ex vivo model of cartilage tissue, a similar response was observed, with palmitate inducing chondrocyte death, IL-6 release, and ECM degradation [85].

Interestingly, the WNT/β-catenin pathway, a known contributor for apoptosis regulation, may be cross-regulated with the TLR/NF-κB signaling pathway [86]. Several WNT pathway components are upregulated in human OA and murine models of exercise-induced OA [87]. Moreover, studies have demonstrated that an imbalance in WNT signaling leads to an OA-like phenotype development in murine models [88]. However, much controversy remains over the role of WNT antagonists, particularly for the role of DKK1 and the expression in OA. WNT antagonists DKK-1 and FRZB mRNA expression levels have been observed to be downregulated in OA cartilage [89]. In sharp contrast, others have shown that DKK1 expression is upregulated during OA progression and correlated with chondrocyte inflammatory markers and apoptosis [90,91,92]. To add further confusion, DKK-1 inhibition or overexpression have both shown protective effects in murine models of OA [91,92]. However, in an in vitro study using human primary chondrocytes extracted from OA patients, DKK1 treatment had a dose-dependent effect increasing chondrocyte apoptosis, mediated IL-1β promotion of chondrocyte apoptosis, in line with the therapeutic hypothesis that attenuating DKK1 may reduce cartilage deterioration in OA [90]. Furthermore, DKK1 KO preventive effects of murine OA model also modulate TLR4 and -9 expression and chondrocyte apoptosis rate [90].

Critically, only a few molecules are known to exert anti-inflammatory protection over chondrocytes, via modulation of TLR-mediated inflammation.

Lubricin, the glycoprotein product of the PRG4 gene, is responsible for boundary-lubrication of the articular cartilage surface, but also has a chondroprotective effect [93]. Lubricin, the main boundary-lubricant of cartilage, can maintain the coefficient of friction at minimal levels, therefore preventing cartilage wear [93]. A vast majority of lubricin is expressed by cartilage superficial zone chondrocytes and synoviocytes, in dimeric and monomeric forms [93]. Surprisingly, synovial fluid resident lubricin has also been shown to down-modulate the TLR2 and TLR4 activation response to PAMPs and OA and RA synovial fluid, leading to reduced inflammation and pain levels in a murine model of OA [94,95]. However, in OA, lubricin synthesis and SF levels are reduced while lubricin proteolytic cleavage is increased, which leads to the loss of lubricin lubricating and anti-inflammatory chondroprotective properties [96]. Consequently, increased chondrocyte apoptosis levels observed in superficial cartilage zones during OA may result from reduced levels of lubricin and its anti-inflammatory and lubricating properties [97].

Another class of molecules associated with anti-inflammatory properties in OA are microRNAs, such as the miRNA-146a and miRNA-146b. MicroRNAs (miRNAs) are small non-coding RNAs that regulate a broad spectrum of physiological cell processes during development and tissue homeostasis via RNA silencing and transcriptional regulation of gene expression. miR-146a and miR-146b have been shown to be upregulated in chondrocytes of OA joints [98,99]. Moreover, miR-146a and miR-146b have also been shown to target pro-inflammatory mediators, particularly those regulating TLR downstream pathways, such as the NF-κB pathway activation in dendritic cells and monocytes [100,101]. Interestingly, a similar mechanism is observed in OA joint chondrocytes. Stimulation with synthetic miR-146 was able to counteract pro-inflammatory molecules, such as IL-1α, by down-regulating induced catabolic molecules [102]. Moreover, mi146a transfection into OA chondrocytes was shown to increase their proliferation and reduce apoptosis by targeting TRAF6 through the NF-kB signaling pathway [103].

## 5. Conclusions

In osteoarthritis, TLRs have a dual capacity to launch both cell defenses and cell death. Rapid progress during the last few years has allowed us to understand the chondrocyte signaling cascades from TLR activation down to cell metabolism; however, the molecular pathways leading to TLR-induced apoptosis remain to be defined. Current data suggest that the latter are as diverse as the former, varying with the TLR type, the joint cell type, and the metabolic reprogramming of the cell. As reviewed above, there are convincing studies on the role of TLR activation and apoptosis of chondrocytes in OA; however, direct evidence demonstrating a causal link between chondrocyte apoptosis and OA remains to be established. Moreover, TLR activation can lead to both pro-survival and pro-apoptotic signals that may have a different relative impact on healthy vs. diseased cells, with an increased tendency to death in osteoarthritic cells. The current findings could also be potentially translated to the clinic, particularly given the potential of soluble TLR4 and others as biomarkers of OA disease severity and progression. However, whether and how those new twists in our understanding of the cross-talk between TLRs and apoptosis could be translated into therapeutic interventions for OA remains to be fully elucidated.

## Figures and Tables

**Figure 1 biology-09-00065-f001:**
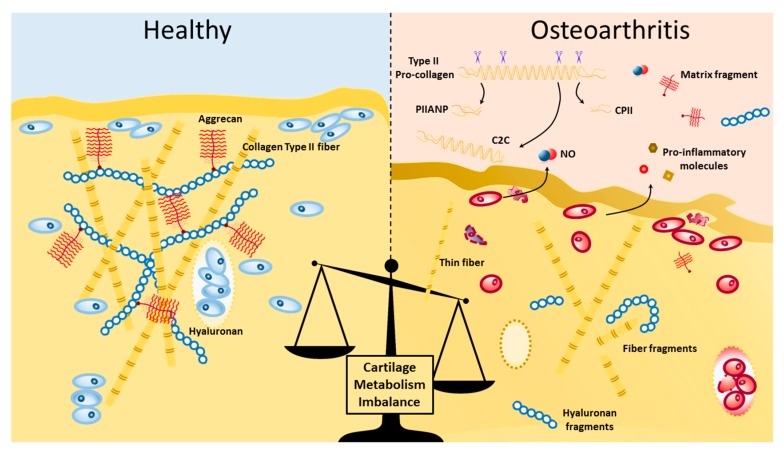
Visual representation of some of the elements of cartilage structure and matrix turnover changes in healthy and osteoarthritic cartilage. Type II collagen fibrils forms an intrinsic network in conjunction with aggrecan aggregates with a hyaluronic acid backbone. In normal physiological conditions, cartilage tissue integrity is maintained by an active turnover of the extracellular matrix, mediated on-site by matrix metalloproteinases (MMPs). In osteoarthritis (OA), there is increased proteolytic damage to matrix molecules in cartilage and remote sites, which can activate receptor signaling pathways, coupled with downregulation of synthesis of collagen and proteoglycan molecules. During OA, chondrocytes also produce and release pro-inflammatory molecules, such as iNOS, interleukin (IL)1, -6, -17, tumor necrosis factor β (TNFβ), interferon (IFN)-alpha, among others.

**Figure 2 biology-09-00065-f002:**
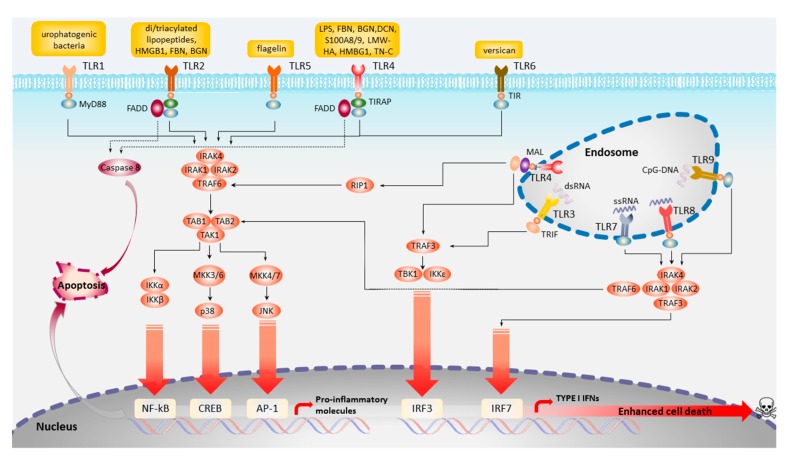
Diagrammatic representation of toll-like receptors (TLR) signaling pathways. AP-1, activator protein 1; BGN, biglycan; CpG-DNA, oligodeoxynucleotides DNA; CREB, cAMP-responsive element-binding protein; DCN, decorin; dsRNA, double-stranded RNA; FADD, FAS-associated death domain; FBN, fibronectin; HMBG1, high-mobility group box 1; IκBα, inhibitor of NF-κBα; IKK, inhibitor of NF-κB kinase; IRAK, interleukin-1 receptor-associated kinase; IRF, IFN-regulatory factor; LPS, lipopolysaccharide; LMW-HA, low molecular-weight hyaluronic acid; MAL, MYD88 adaptor-like protein; MD2, myeloid differentiation factor 2; MEK, mitogen-activated protein kinase/ERK kinase; MKK, mitogen-activated protein kinase; NF-κB, nuclear factor-κB; PI3K, phosphoinositide 3-kinase; RIP1, receptor-interacting protein 1; S100A8/9, S100 calcium-binding protein A 8 and 9; ssRNA, single-stranded RNA; TAB, TAK1-binding protein; TAK1, TGFβ-activated kinase 1; TN-C, tenascin-c; TRAF, tumor necrosis factor receptor-associated factor; TRAM, TRIF-related adaptor molecule.

**Table 1 biology-09-00065-t001:** Toll-like receptors and their respective endogenous ligands present in osteoarthritis.

Receptor	Receptor Expression	DAMPs	Modulators	Reference
TLR1/2	FLS, chondrocytes	N/A	N/A	[18]
TLR2	FLS, chondrocytes	HMGB1, Fibronectin fragments	Mechanical strain, TNF-α	[57,58]
TLR3	FLS, chondrocytes	dsRNA	N/A	[18,59]
TLR4	FLS, chondrocytes	Fibronectin fragments, S100A8-A9, TN-C, LMW-HA, HMGB1, biglycan, decorin	Mechanical stress, TNF-α, plasma proteins	[58]
TLR5	FLS, chondrocytes, bone	Functionally proofed but ligand unidentified	TNF-α, IL-8	[60]
TLR6/2	FLS, chondrocytes	Versican	N/A	[61,62]
TLR7	FLS, chondrocytes	ssRNA	N/A	[18,59]
TLR8	FLS, chondrocytes	ssRNA	N/A	[18]
TLR9	FLS, chondrocytes	dsRNA	N/A	[18]

ssRNA: single-stranded RNA; dsRNA: double-stranded RNA; FLS: fibroblast-like synoviocytes; TN-C: tenascin-c; SF: synovial fluid; LMH-HA: low-molecular-weight hyaluronan; HMGB1: high-mobility group box 1; TNF-α: tumor necrosis factor alpha; IL-6: interleukin 6; IL-8: interleukin 8; S100A9: S100 calcium-binding protein A9.

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
