# Peer review of "Osteoarthritis and Toll-Like Receptors: When Innate Immunity Meets Chondrocyte Apoptosis"

_biology, 2020, doi:10.3390/biology9040065_

Round 1
Reviewer 1 Report
In this review article, the authors provide an overview of the TLR-induced innate immune responses during osteoarthritis development setting the focus on articular chondrocyte apoptosis.
In general, the topic is relevant and interesting, however, there are some points that need improvement and clarification.
- The authors state in the abstract that “Apoptosis is of particular importance, given that chondrocyte death is a dominant feature in OA“ and the title „1. Introduction: The Role of Inflammation and Apoptosis in OA“ announces that information on chondrocyte apotosis will be provided in this section. This information is missing. The authors should provide a passage on the importance of chondrocyte apoptosis in OA pathogenesis (difference to necrosis regarding TLR-mediated effects ….?).
- Line 41: „resident tissues“ should be resident cells and „matrices“ matrix
- Section 2, lines 52-70: Please, provide references for passage 1, 2 and 3.
- Section 2, line 59: be careful with wording. „The enzymatic degradation of articular cartilage IS the erosion of pericellular matrix and eventually the interterritorial matrix“
- Section 2, line 71: First, the authors describe that cartilage degradation is prominent in the superficial zone and pericellular matrix (which is present in all zones….). Then, in the next sentence, they wrote that at this zone and throughout the osteoarthritic cartilage MMPs etc are associated with matrix degradation. This is confusing. What happens where?
- Section 2, lines 79-85: in my opinion, this general information should come earlier, especially because healthy situation is described. As next, OA situation with disturbed anabolic-catabolic balance could follow…
- Fig.1: Please define the abbreviations CPII, C2C, NO, PIIANP in the legende.
- Fig. 2: does apoptosis (induced by NFkB, left) mean the same as type 1 IFN-induced cell death (right)?
- Fig.2: Please define HMBG1, FBN, BGN, CpG-DNA etc. in the legende.
- Line 202: reference error
- Table1: I miss the reference to Table 1 in the manuscript text.
- Section 4: TLR and Chondrocyte Apoptosis would be more clear
Reviewer 2 Report
Biology-741529
Comments to the authors
This review by Barreto, G., et al., is an interesting and necessary review on osteoarthritis and TLRs, with particular emphasis on the role of TLR signaling in chondrocyte apoptosis. However, the authors should also mention 2 recent reviews about the same topic or quite similar:
- Danger signals and inflammaging in osteoarthritis by Millerand, M., et al. Exp. Rheumatol. 2019; 37 (Suppl. 120): S48-S56.
- Inflammation in osteoarthritis: is it time to dampen the alarm(in) in this debilitating disease? Exp. Immunol., 2018; 195: 153–166.
There are some editing errors along the manuscript that should carefully revised by the authors. Thus, in line 202, the phrase “Error! Reference source not found.1” has no sense at all. My guess is that should be changed by “Table 1” instead.
In line 232, the b of “TGF-b-activated kinase 1” should be substituted by the Greek symbol β, or spelled out as “TGF-beta-activated kinase 1”
In line 328, the same for the a in “IL-1a”, it should be IL-1α
Round 2
Reviewer 1 Report
The authors did respond to all of my questions suggestions.